# Modeling the Overdispersion of *Pasteuria penetrans* Endospores

**Ioannis Vagelas** [1,*] and **Stefanos Leontopoulos** [2]

1    Department of Agriculture Crop Production and Rural Development, University of Thessaly, Fytokou Str., Nea Ionia, GR-38446 Magnesia, Greece

2    Department of Agricultural Engineering Technologists, School of Agricultural Technology, University of Thessaly, Gaiopolis, GR-41500 Larissa, Greece; sleontopoulos@uth.gr

\*    Correspondence: vagelas@uth.gr; Tel.: +30-69-7829-9170

**Abstract:** This paper discusses a process of developing the data analysis and modeling of *Pasteuria penetrans* spore attachment in vitro and in planta, based on the observation that the number of spores attaching to juveniles within a given time increased by increasing the time of exposure to spores and the spores dose. Based on this, the *P. penetrans* spore attachment in vitro was modeled using the negative binomial distribution which permits decomposing the observation's variability into three components: randomness, internal differences between individuals, and the presence of other external factors, e.g., the soil type. Additionally, we developed case-detection methods to explain *P. penetrans* spores' attachment variability. The statistical methods developed in this paper show that a nematodes invasion is significant limited when second stage juveniles (J2s) are encumbered with seven *P. penetrans* spores. This research study concludes that the number of spores attached in J2s, the time of exposure of J2s to *P. penetrans* spores, and the soil texture are important factors affecting the invasion of root-knot nematodes in tomato plants.

**Keywords:** biological control; *Meloidogyne* spp.; probability models; negative binomial distribution; Poisson distribution; Weibull and gamma (γ) distributions





## 1. Introduction

Plant-parasitic nematodes are recognized as major agricultural pathogens and are known to attack plants and cause great economic losses in crops worldwide [1–3]. Among plant-parasitic nematodes, there are sedentary endoparasites, which include the genera *Heterodera* and *Globodera* (cyst nematodes) and *Meloidogyne*, which are species better known as root-knot nematodes. Vegetables and ornamental crops are usually among the most susceptible cultivated plants that are affected by root-knot nematodes [4]. Among root-knot nematodes, *Meloidogyne javanica* is one of the most damaging plant parasites, often causing heavy losses, especially in plants grown in coarse-textured sandy soils. Nowadays, alternatives including the use of chemical substances, agricultural practices for managing pests, and the disease incorporation of organic amendments are either environmentally unacceptable or technically demanding. Thus, resistant varieties may potentially provide the most effective means of controlling nematode invasion. Despite the aforementioned control methods, the use of biological control agents such as the obligate hyperparasitic bacterium *Pasteuria penetrans* [5–7] has been recognized as one of the most effective biological control agents against root-knot nematodes [8,9], showing quite promising results [10–12]. The potential use of *P. penetrans* as a biological control agent against important plant-parasitic species has been widely studied in the past [13–16]. These studies include the distribution rate, host range, host specificity, as well as biotic and abiotic factors [17–20]. To date, it has been mentioned that successful parasitism depends on the attachment of 5–10 spores per juvenile (J2), which is sufficient to initiate infection without reducing the ability of the nematode to invade the root system [21–23]. However, studies have shown that there may be little or no root invasion if there are more than 15 spores attached to the nematode's

body, inferring that spore attachment will affect the ability of J2s to locate and/or invade the root [24–26]. The results of these studies imply high variances in the numbers of spores attached to the nematode's body but no attempt has been made to examine this variability in detail, e.g., as shown by the target parameter of variability. This research is important since plant protectionists could use it as a prediction model to decide whether infected nematodes can cause plant damage and thus, yield losses.

Studies focused on the biology of *P. penetrans* mentioned that, during the infective stage, the endospores of *P. penetrans* attach to the extremal nematode body wall (cuticle) of free-living of *Meloidogyne* populations. As observed in many relevant studies [27,28], as soon as J2s are attracted by the root exudates of a host plant, the nematodes begin to become attached by the bacterium spores which penetrate the nematode's cuticle and begin to grow and develop within the invaded nematode [6,29,30]. Eventually, the female nematode body becomes filled with spores [10,30]. As it has been remarkably noted, each infected female may contain up to 2.5 million spores [31], which are finally released back into the soil environment.

Overall, in this study, the main objective was to calculate the *P. penetrans* spore attachment process as a distribution, providing a model that allows the variance to be divided into components and offer an explanation in vitro and in planta.

## 2. Results

### 2.1. Modeling P. penetrans Spore Attachment Data—Water Bioassay

The results of our study of *P. penetrans* spores' attachment show that a J2 of root-knot nematode may be encumbered with one or more spores over a fixed period (Figure 1). If every J2 were equally exposed to the chance of being encumbered with *P. penetrans* spores over a fixed period, the distribution would follow the Poisson series and the expected variance ($s^2$) is equal to a mean. As Table 1 shows, the observed variance ($s^2$) is significantly larger than the mean at 3, 6, and 9 h of incubation. Therefore, successful events of the attachment of *P. penetrans* to the nematode cuticle cannot be formulated with a Poisson process as the parameter $s^2$ is not smaller than or equal to the mean.

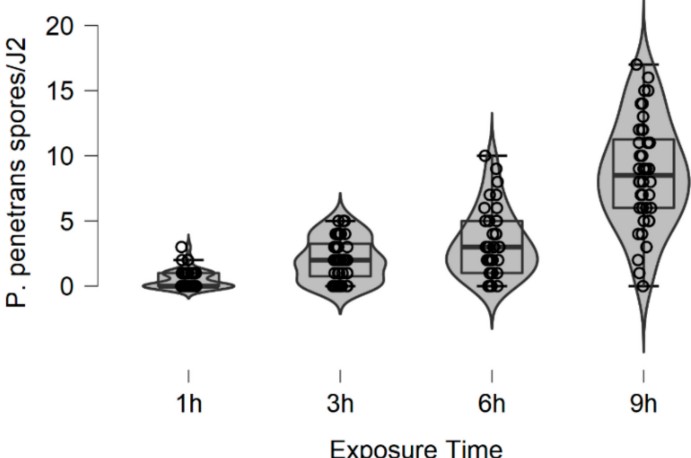

**Figure 1.** JASP output for *P. penetrans* spores' attachment. Boxplots with violins and jittered data are shown separately for the four exposure times of the estimates for the attachment of *P. penetrans* to J2s.

As shown in Table 1, mean numbers, (e.g., at 9 h of nematode exposure to a *P. penetrans* spore suspension), indicating strong overdispersion. This suggests that *P. penetrans* spores are clumped, and more than one spore sticks on the cuticle of each J2. This observation agrees with relevant research studies wherein it was indicated that the 'overdispersion' phenomenon is common for living organisms [32–34] such as plant-parasitic nematodes [24,26,35] and can be explained by the negative binomial distribution.

**Table 1.** Descriptive statistics of *P. penetrans* spores' attachment at different times of exposure.

| | *P. penetrans* Spores/J2 | | | |
|---|---|---|---|---|
| | **1 h** | **3 h** | **6 h** | **9 h** |
| Sample size (*n*) | 40 | 40 | 40 | 40 |
| Mean | 0.513 | 2.025 | 3.275 | 8.600 |
| Std. deviation | 0.721 | 1.641 | 2.522 | 4.131 |
| Variance ($s^2$) | 0.520 | 2.692 | 6.358 | 17.067 |

Based on these results, data show that a better fit is obtained with the negative binomial distribution for *P. penetrans* spore attachment per juvenile at 3, 6, and 9 h after application (Figure 2). According to the data presented in previous research work [24,26], in all cases (3, 6, and 9 h), it was observed that the negative binomial distribution proved to be a better model for predicting the observed values compared with the Poisson distribution (Figure 3). The chi-square test of the hypothesis shows that the negative binomial (NegBin) model was the most appropriate to fit the observed counts (Table 1).

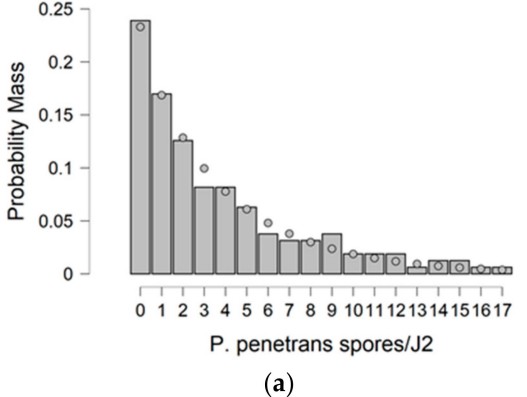

(**a**)

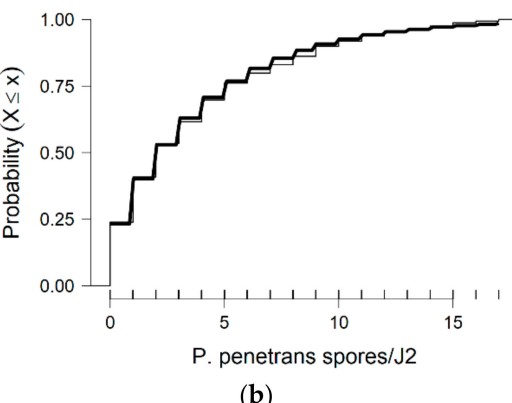

(**b**)

**Figure 2.** Negative binomial distribution plots for *P. penetrans* spores' attachment: (**a**) probability mass function (PMF); and (**b**) cumulative distribution function (CDF). (**a**) displays a histogram of the selected variable overlaid with the probability density function of the fitted distribution (dots). (**b**) displays an empirical cumulative distribution plot overlaid with the cumulative distribution function of the fitted distribution (solid line).

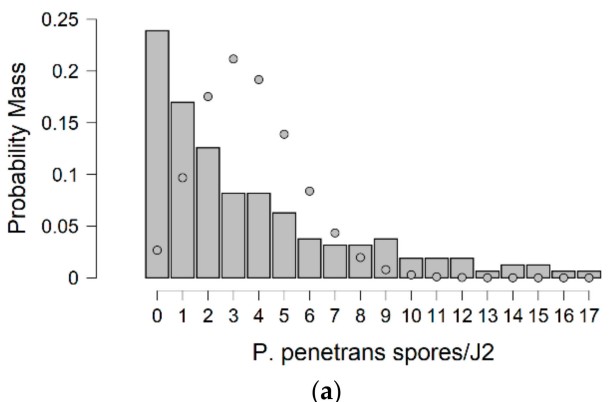

(**a**)

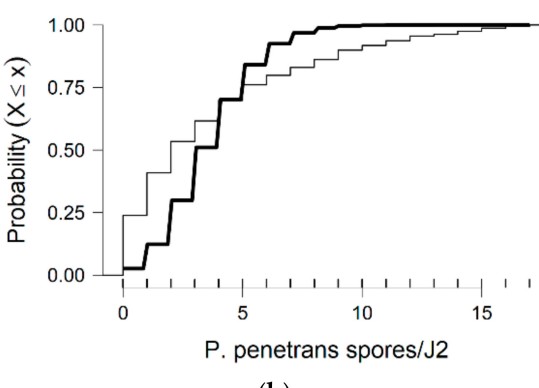

(**b**)

**Figure 3.** Poisson distribution plots for *P. penetrans* spores' attachment: (**a**) probability mass function (PMF); and (**b**) cumulative distribution function (CDF). (**a**) displays a histogram of the selected variable overlaid with the probability density function of the fitted distribution (dots). (**b**) displays an empirical cumulative distribution plot overlaid with the cumulative distribution function of the fitted distribution (solid line).

Moreover, the results presented in Figure 2 show that the negative binomial is the most appropriate distribution, mostly fitting all observations. This could be explained due to (a) the observed variance ($s^2$) which is larger than the mean (Table 1); and (b) as time increases, the overdispersion becomes excessively large for the Poisson distribution (Table 2).

**Table 2.** Estimate of the best fit probability for NB and Poisson distribution.

| Negative Binomial (NB) Distribution | | | Poisson Distribution | | |
|---|---|---|---|---|---|
| **Test** | **Statistic** | ***p*** | **Test** | **Statistic** | ***p*** |
| Chi-square | 5.382 | 0.996 | Chi-square | 38,827.259 | <0.001 |

### 2.2. Estimating Root-Knot Nematodes Development by Water Bioassay

Results of the study of *P. penetrans* spores' attachment show that the means at 9 and 36 h of nematode exposure to *P. penetrans* spore suspension are smaller (Table 3), e.g., equal to half the variance and thus indicating strong overdispersion. The J2s attachment range was 8.6 spores per J2 after 9 h and 28.2 spores per J2 after 36 h (Table 3). The 25th and 75th percentile provide an accurate picture of the attachment boundaries, e.g., since 6–11.2 spores were attached per J2 after 9 h (Table 3), meaning that half of the nematodes achieved either above or below the 50th percentile (median), which is equal to 8.5 (Table 3). These are the same values plotted in boxplots with violins and jittered data (Figure 4). In both cases (9 and 36 h), it was observed that the negative binomial distribution proved to be a better model for predicting the observed values compared with the Poisson distribution. Figure 5 indicates that, after a long period, the nematode only encounters clumps of spores. In a 36 h bioassay (Table 3), the most suppressive effect was achieved by J2s encumbered with 24 - 32 *P. penetrans* spores (25–75th percentile). Based on this observation, when J2s were encumbered with 24–32 spores of *P. penetrans*, these nematodes could not reset the gall and egg mass (Figures 6 and 7), suggesting that *P. penetrans* spores probably halt J2s invasion when high numbers of spores are attached to the nematode's cuticle. In the 9 h bioassay, when J2s were encumbered with 6–11 *P. penetrans* spores (Table 3, 25–75th percentile), and the nematodes produced significantly higher values of gall and egg masses compared to the 36 h attachment data (Figures 6 and 7), suggesting that the *P. penetrans* spores' density affects the root-knot invasion.

**Table 3.** Descriptive statistics of *P. penetrans* spores' attachment at different times of exposure.

| | *P. penetrans* Spores/J2 | |
|---|---|---|
| | **36 h** | **9 h** |
| Sample size (*n*) | 40 | 40 |
| Mean | 28.2 | 8.6 |
| Std. deviation | 6.5 | 4.1 |
| Variance | 42.6 | 17.1 |
| 25th percentile | 24.0 | 6.0 |
| 50th percentile | 27.5 | 8.5 |
| 75th percentile | 32.0 | 11.2 |

Figure 8 demonstrates the relationship between the nematodes gall and egg masses. More specifically, in Figure 8, the "Ghost Line" from the middle panel (J2s treated with 6–11 *P. penetrans* spores) is repeated in red across the other two panels. In more detail, the "Ghost Line" clearly suggests that the relationship between the gall and the egg masses differs among the panels. With the "Ghost Line," it can be easily recognized that J2s encumbered with high numbers (24–32) of *P. penetrans* spores cannot enter plant tissue, probably due to their immobility in the soil.

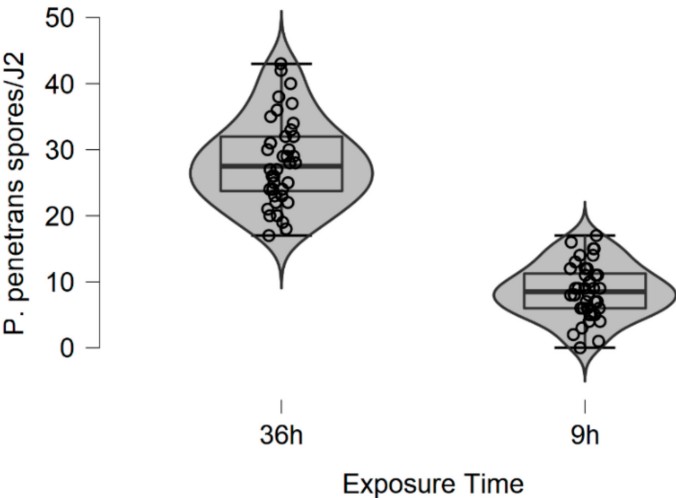

**Figure 4.** JASP output for *P. penetrans* spores' attachment. Boxplots with violins and jittered data are separately shown for the four exposure times of the estimates for the attachment of *P. penetrans* to J2s.

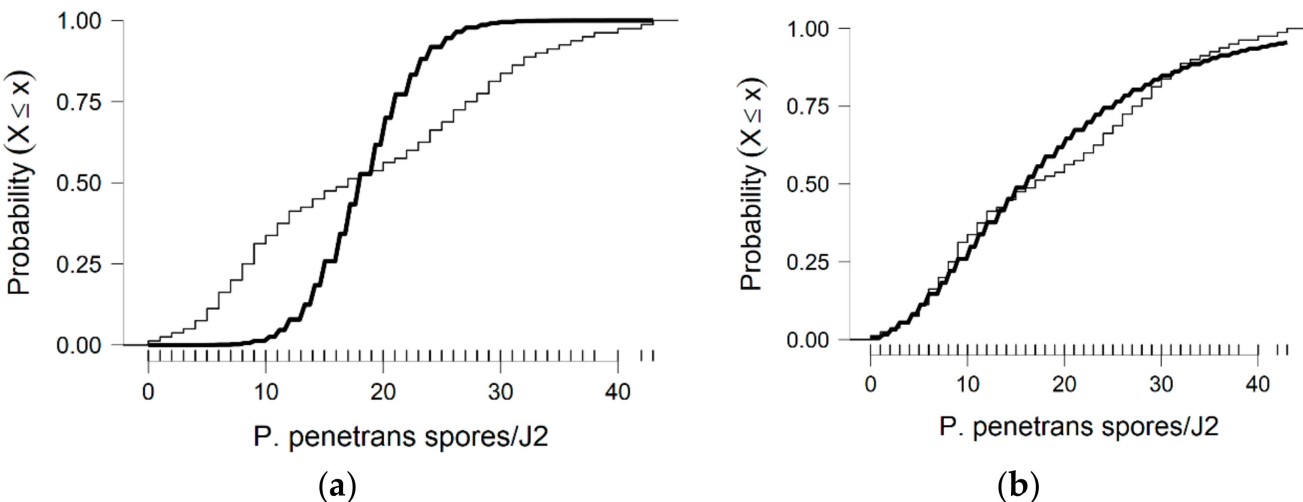

**Figure 5.** Cumulative distribution function (CDF) for Poisson and negative binomial distribution for *P. penetrans* spores' attachment, where (**a**) the CDF for Poisson distribution and (**b**) the CDF for negative binomial distribution.

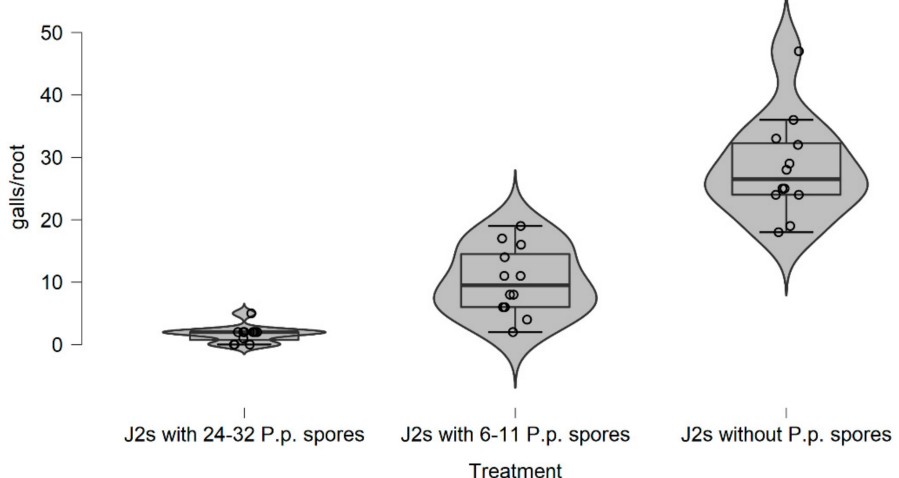

**Figure 6.** Effect of *P. penetrans* (P.p.) spores' attachment on gall formation when J2s were encumbered with 24–32 or 6–11 P.p. spores at exposure times of 36 and 9 h, respectively.

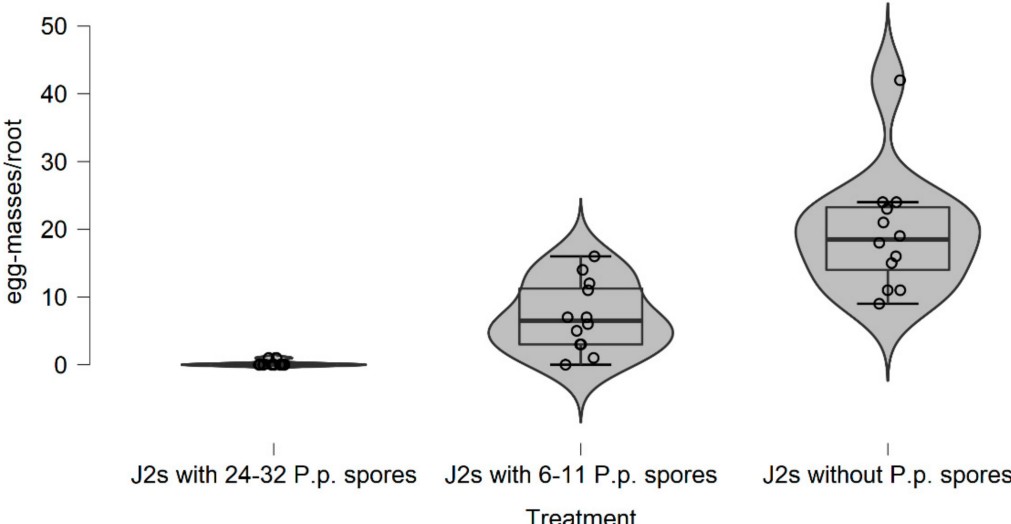

**Figure 7.** Effect of *P. penetrans* (P.p.) spores' attachment on egg mass formation when J2s were encumbered with 24–32 or 6–11 P.p. spores at exposure times of 36 and 9 h, respectively.

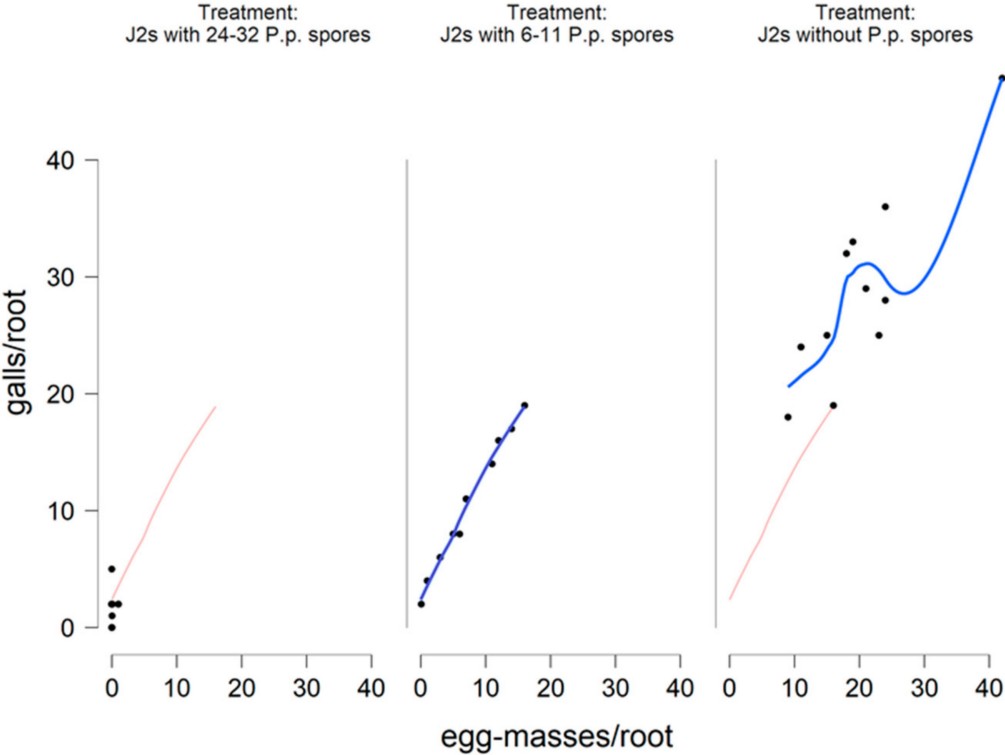

**Figure 8.** Plots (panels) reflecting the effect of *P. penetrans* (P.p.) spores' attachment on egg mass formation when J2s were encumbered with 24–32 or 6–11 P.p. spores at exposure times of 36 and 9 h, respectively. "Ghost Lines" (red lines) repeat the pattern from one to the others, making it easier to compare the differences between the panels.

*2.3. Modeling P. penetrans Spore Attachment Data—Soil Bioassay*

The data presented in Figure 9 show that, in a sandy loam bioassay for *P. penetrans* spore attachment at 5000 and 15,000 *P. penetrans* spores/J2, the best fit was obtained with the negative binomial (Figure 10), and not with the Poisson distribution (Figure 11).

The Poisson distribution is assumed for modeling for loam bioassay (Table 4). The data presented in Figure 12 show that, for the loam bioassay, the best fit is obtained with both the Poisson and the negative binomial distributions (Figures 13 and 14), suggesting

that the *P. penetrans* spore is not attached in clumps or a failure has occurred due to the soil stress environment.

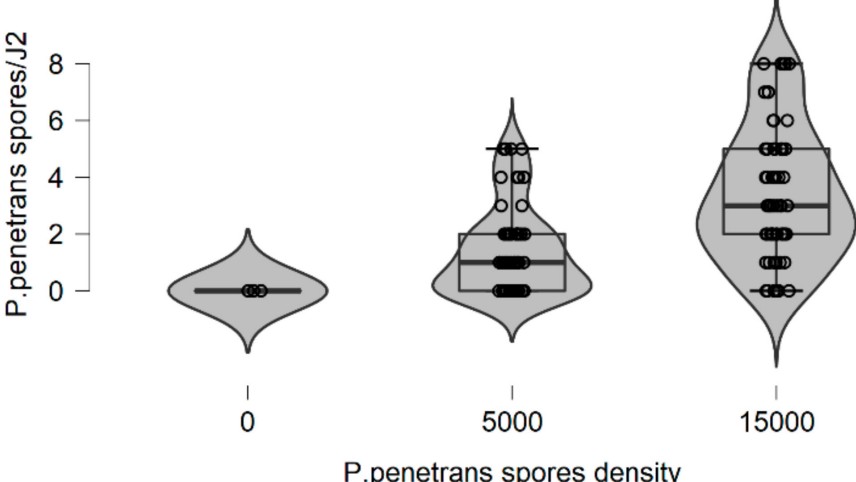

**Figure 9.** JASP (open source statistical software) output for *P. penetrans* spores' attachment. Boxplots with violins and jittered data are shown separately for the *P. penetrans* spores' densities such as 0, 5000 and 15,000 *P. penetrans* spores/mL in sandy loam soil bioassays.

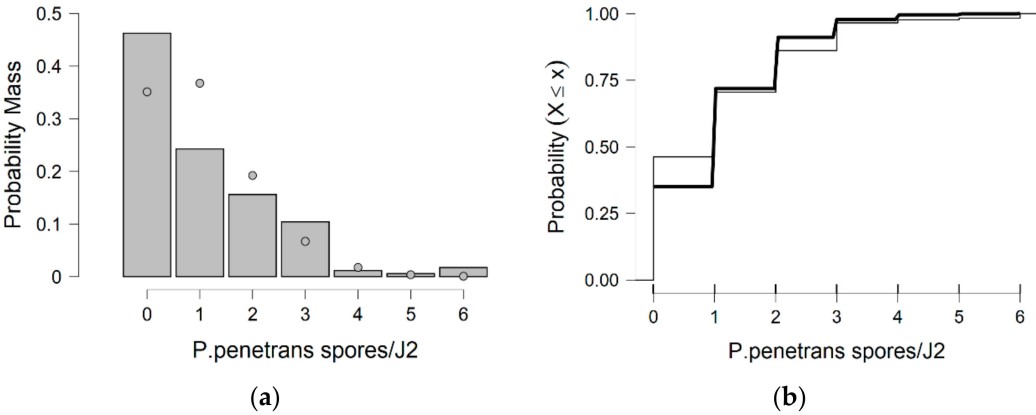

(**a**)            (**b**)

**Figure 10.** Negative binomial distribution plots for *P. penetrans* spores' attachment: (**a**) probability mass function (PMF) and (**b**) cumulative distribution function (CDF). (**a**) displays a histogram of the selected variable overlayed with the probability density function of the fitted distribution (dots). (**b**) displays an empirical cumulative distribution plot overlayed with the cumulative distribution function of the fitted distribution (solid line).

**Table 4.** Loam bioassay: estimate of the best fit probability for NB and Poisson distribution.

| Negative Binomial (NB) Distribution | | | Poisson Distribution | | |
|---|---|---|---|---|---|
| Test | Statistic | *p* | Test | Statistic | *p* |
| Chi-square | 10.708 | 0.098 | Chi-square | 94.087 | <0.001 |

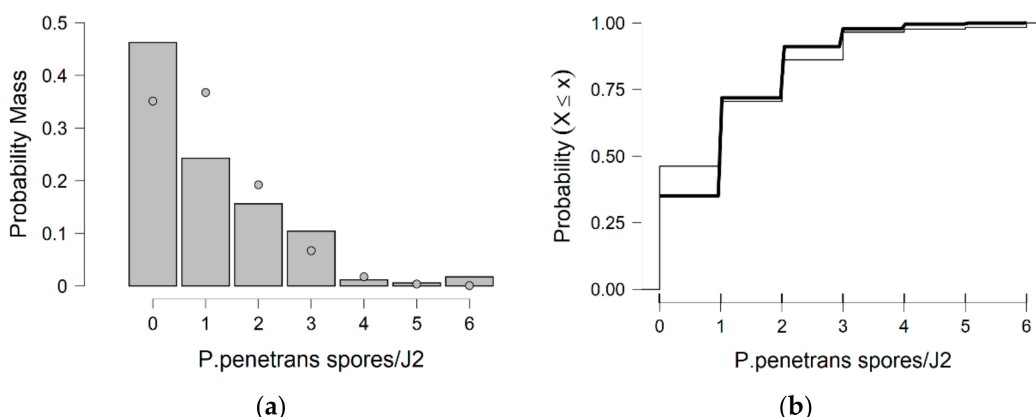

(**a**)                                                              (**b**)

**Figure 11.** Poisson distribution plots for *P. penetrans* spores' attachment: (**a**) probability mass function (PMF) and (**b**) cumulative distribution function (CDF). (**a**) displays a histogram of the selected variable overlayed with the probability density function of the fitted distribution (dots). (**b**) displays an empirical cumulative distribution plot overlayed with the cumulative distribution function of the fitted distribution (solid line).

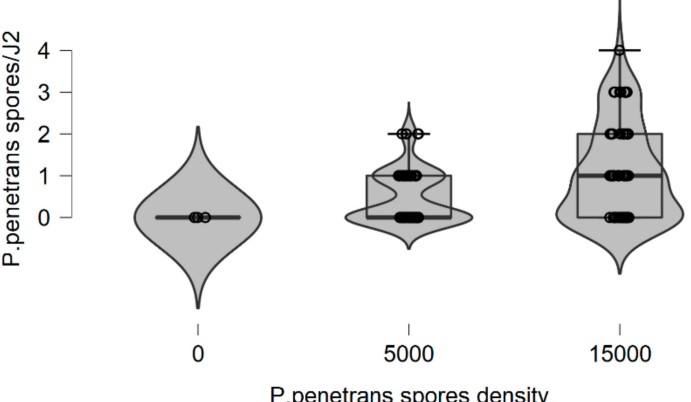

**Figure 12.** JASP output for *P. penetrans* spores' attachment. Boxplots with violins and jittered data are shown separately for the four *P. penetrans* spores' densities, namely 0, 500, 5000, and 15,000 *P. penetrans* spores/mL in loam soil bioassays.

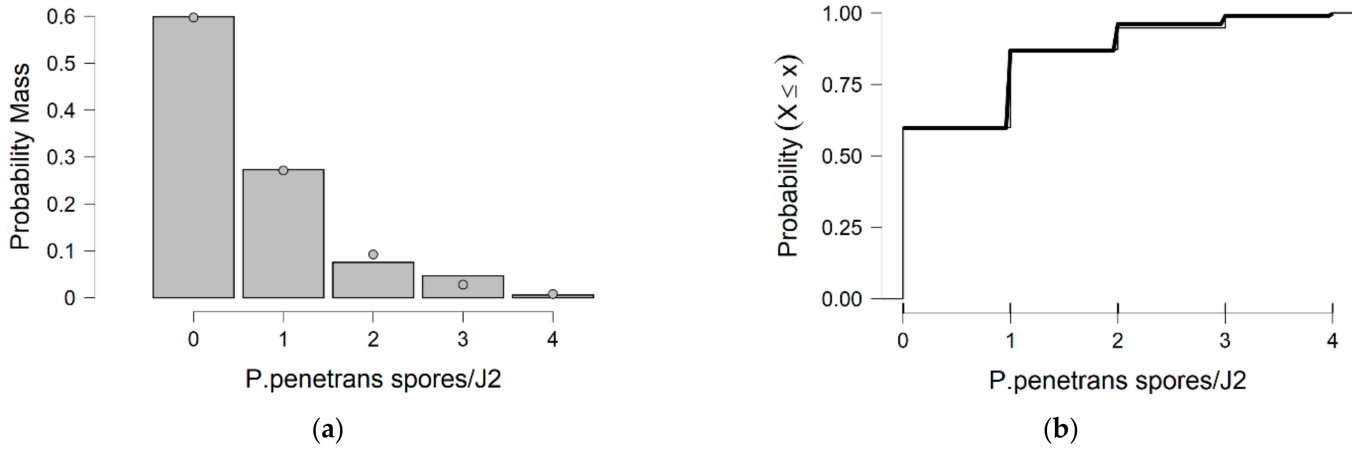

(**a**)                                                              (**b**)

**Figure 13.** Negative binomial distribution plots for *P. penetrans* spores' attachment: (**a**) probability mass function (PMF) and (**b**) cumulative distribution function (CDF). (**a**) displays a histogram of the selected variable overlayed with the probability density function of the fitted distribution (dots). (**b**) displays an empirical cumulative distribution plot overlayed with the cumulative distribution function of the fitted distribution (solid line).

Figures 10 and 11 showed that the negative binomial is the more appropriate distribution fitting all observations only for sandy loam bioassay treatment (Table 5). A possible explanation could be that (i) as time increases, the overdispersion was excessively large for sandy loam bioassay, whereas (ii) the Poisson distribution (Figures 13 and 14) is also considered an appropriate model to fit the loam bioassay data, suggesting an 'underdispersion' or a failure of *P. penetrans* spores in the attachment ability due the stress environment which occurred in the loam soil.

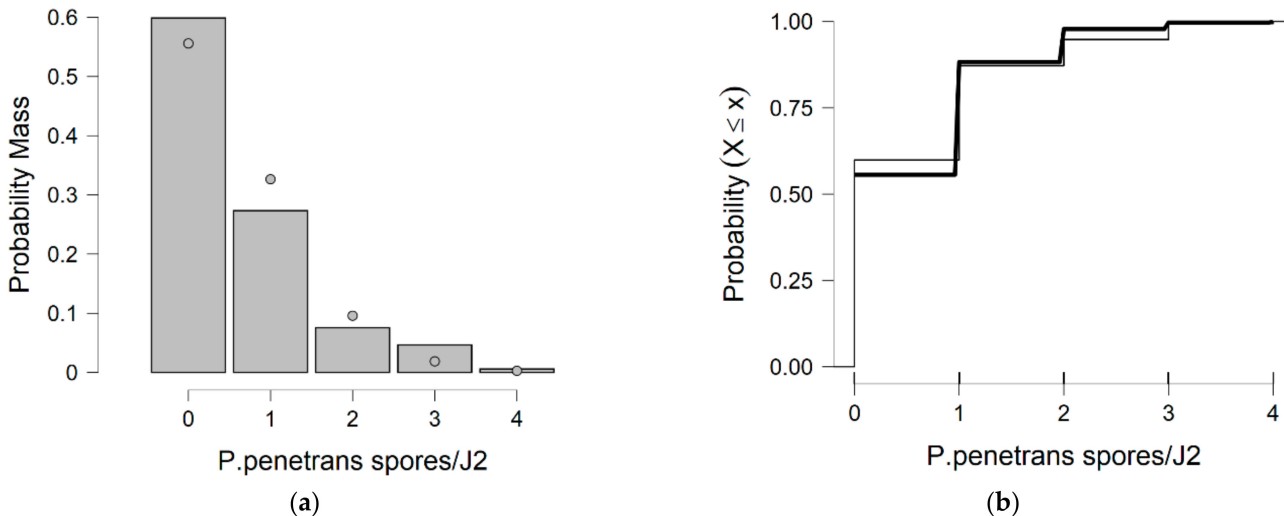

(**a**)                                    (**b**)

**Figure 14.** Poisson distribution plots for *P. penetrans* spores' attachment: (**a**) probability mass function (PMF) and (**b**) cumulative distribution function (CDF). (**a**) displays a histogram of the selected variable overlayed with the probability density function of the fitted distribution (dots). (**b**) displays an empirical cumulative distribution plot overlayed with the cumulative distribution function of the fitted distribution (solid line).

**Table 5.** The estimate of the best fit probability for NB and Poisson distribution.

| Negative Binomial (NB) Distribution | | | Poisson Distribution | | |
| --- | --- | --- | --- | --- | --- |
| Test | Statistic | *p* | Test | Statistic | *p* |
| Chi-square | 2.741 | 0.602 | Chi-square | 10.439 | 0.034 |

*2.4. Estimainge J2s Root Invasion Rate Encumbered with Different Doses of P. penetrans Spores in Sandy Loam and Loam Soil*

The data presented in Figure 15 show that (i) less J2s invaded the plants roots when encumbered with high rates of *P. penetrans* spores, e.g., 15,000 spores/mL; (ii) and the soil type attract different rates of J2s to invade the root system. For J2s' invasion, soil type and *P. penetrans* spores have a significant impact (Table 6). These results suggest that J2s do not differentiate in terms of the invasion in the sandy loam soil (when treated with 5000 Pp spores'/mL) compared with the J2s invasion in loam soil treated with 5000 or 15,000 Pp spores'/mL (Figure 15).

**Table 6.** ANOVA table for J2s' invasion encumbered in treatments with different dose densities of *P. penetrans* spores and two soil types (sandy loam and loam soil).

| Cases | Sum of Squares | df | Mean Square | F | *p* |
| --- | --- | --- | --- | --- | --- |
| Soil type | 2106.750 | 1 | 2106.750 | 55.780 | <0.001 |
| *P. penetrans* spores density | 2945.333 | 1 | 2945.333 | 77.983 | <0.001 |
| Soil type *P. penetrans* spores density | 14.083 | 1 | 14.083 | 0.373 | 0.545 |
| Residuals | 1661.833 | 44 | 37.769 | | |

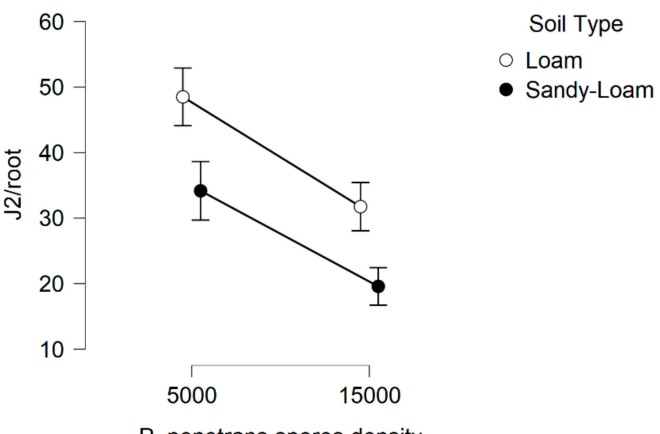

**Figure 15.** Descriptive plot for the J2s' invasion in treatments with different dose densities of *P. penetrans* spores and two soil types, namely sandy loam and loam soil. Error bars represent 95% confidence interval.

The data presented in Figure 16 show that (i) attachment is influenced by the soil type and *P. penetrans* spore density; and (ii) the nematodes in sandy loam treated with 15,000 Pp spores/mL were three times more attached with *P. penetrans* spores/mL compared with those (J2s) in the loam soil treated with 15,000 Pp spores/mL. Soil type has a significant impact on the ability of *P. penetrans* spores to attach the J2s cuticle. (Table 7), suggesting that environmental factors such as soil texture influence the J2s invasion when they are encumbered with *P. penetrans* spores.

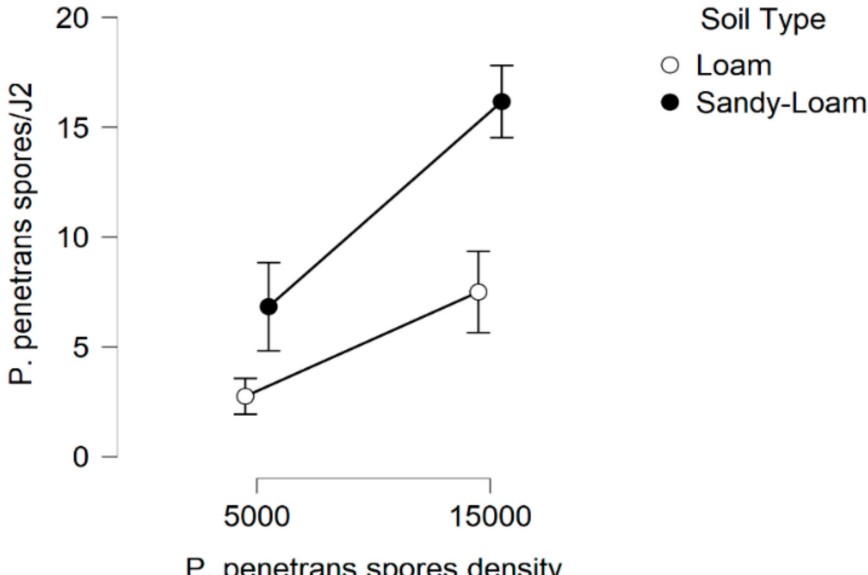

**Figure 16.** Descriptive plot for J2s' attachment in treatments with different dose densities of *P. penetrans* spores and two soil types, namely sandy loam and loam soil. Error bars represent 95% confidence interval.

In the study of J2s' invasion, fewer nematodes invade roots when they are treated with 15,000 *P. penetrans* spores/mL (Figures 15 and 17). The nematodes invasion was significantly reduced when the J2s were encumbered with more than 6 *P. penetrans* spores (Figure 17, blue line). The "Ghost Line" in Figures 17 and 18 represents this influence over J2s invasion when J2s are encumbered with more than six spores of *P. penetrans*.

**Table 7.** ANOVA table for J2s' attachment in treatments with different dose densities of *P. penetrans* spores and two soil types (sandy loam and loam soil).

| Cases | Sum of Squares | df | Mean Square | F | *p* |
|---|---|---|---|---|---|
| Soil type | 487.687 | 1 | 487.687 | 72.843 | <0.001 |
| *P. penetrans* spores density | 595.021 | 1 | 595.021 | 88.874 | <0.001 |
| Soil type *P. penetrans* spores density | 63.021 | 1 | 63.021 | 9.413 | 0.004 |
| Residuals | 294.583 | 44 | 6.695 | | |

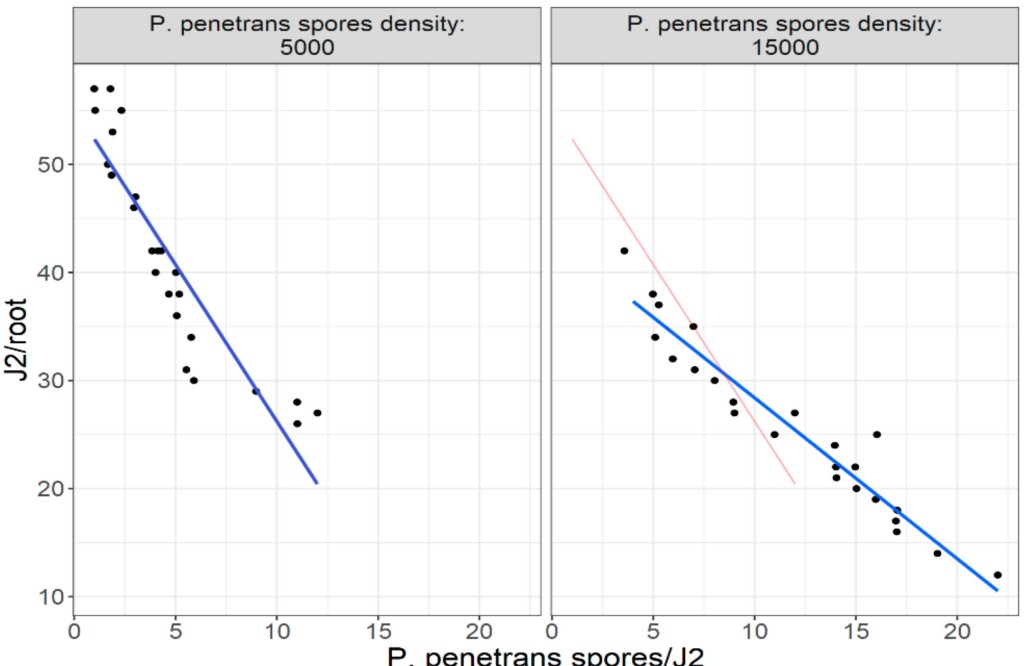

**Figure 17.** Plots (panels) reflecting the relationship between the nematodes invasion (J2s/root) and J2s encumbered with *P. penetrans* spores (*P. penetrans* spores/J2) when treated with 5000 or 15,000 Pp spores/mL. "Ghost Line" (red line) repeats the pattern from one to the others, making it easier to compare the differences among panels.

The data presented in Figure 17 show that even the lowest *P. penetrans* concentration, (5000 *P. penetrans* spores/mL) significantly reduced the J2s invasion when nematodes were encumbered with >7 *P. penetrans* spores (blue arrows). At the highest *P. penetrans* concentration (15,000 *P. penetrans* spores/mL), a lower rate of nematode invasion was observed, particularly when J2s were encumbered with >15 *P. penetrans* spores (Figure 17, red line). In order to better understand how each of the levels of the factors (soil type and *P. penetrans* spores' density) related to the number of *P. penetrans* spores attached to J2 we examined, the ordinary linear regression using the Minitab statistical program. The positive coefficient of 6.375 for sandy loam soil (Table 8) indicates that more *P. penetrans* spores became attached in sandy loam soil compared to in loam soil. The coefficient value for the Poisson regression model fit is very small (0.808, Table 8), indicating poor model fit. In fact, the Poisson model estimates were more erratic due to the overdispersion. This was observed as the variance of the fitted values were nearly two times greater than the mean (Table 9).

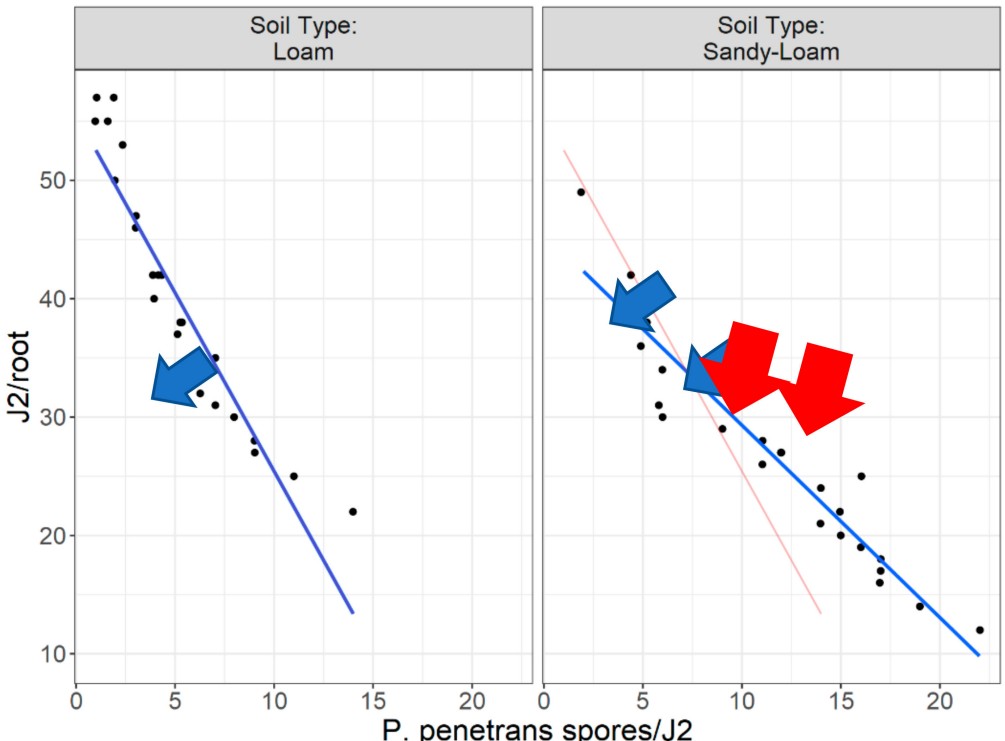

**Figure 18.** Plots (panels) reflecting the relationship between the nematodes invasion (J2s/root) and J2s encumbered with *P. penetrans* spores (*P. penetrans* spores/J2) in two soil types (sandy loam and loam soil). "Ghost Line" (red line) repeats the pattern from one to the others, making it easier to compare the difference between panels.

**Table 8.** Coefficients table (regression analysis) of *P. penetrans* spores'/J2 vs. *P. penetrans* spores' density; soil type.

| Coefficients Term | Ordinary Linear Regression | | | | | Poisson Regression | | | | |
|---|---|---|---|---|---|---|---|---|---|---|
| | Coef | SE Coef | T-Value | *p*-Value | VIF | Coef | SE Coef | Z-Value | *p*-Value | VIF |
| Constant | −1.917 | 0.997 | −1.92 | 0.061 | | 0.631 | 0.161 | 3.91 | 0.000 | |
| *P. penetrans* spores density | 0.000704 | 0.000081 | 8.65 | 0.000 | 1.00 | 0.000090 | 0.000011 | 8.18 | 0.000 | 1.00 |
| Soil type | | | | | | | | | | |
| Sandy loam | 6.375 | 0.814 | 7.83 | 0.000 | 1.00 | 0.808 | 0.108 | 7.46 | 0.000 | 1.00 |

**Table 9.** Descriptive statistics of (J2s/root) and J2s encumbered with *P. penetrans* spores (P.p spores density) in two soil types (sandy loam and loam soil).

| | J2/Root | | | | J2s Encumbered with *P. penetrans* Spores | | | |
|---|---|---|---|---|---|---|---|---|
| | P.p Spores' Density | | Soil Type | | P.p Spores' Density | | Soil Type | |
| | 5000 | 15,000 | Loam | Sandy Loam | 5000 | 15,000 | Loam | Sandy Loam |
| Sample size (*n*) | 24 | 24 | 24 | 24 | 24 | 24 | 24 | 24 |
| Mean | 41.333 | 25.667 | 40.125 | 26.875 | 4.792 | 11.833 | 5.125 | 11.500 |
| Std. deviation | 10.007 | 8.020 | 10.588 | 9.424 | 3.148 | 5.181 | 3.275 | 5.540 |
| Variance | 100.145 | 64.319 | 112.114 | 88.810 | 9.911 | 26.841 | 10.723 | 30.696 |

The data presented in Figure 18 show that, in the two examined soil types, fewer nematodes invade plant roots when they are encumbered with >7 *P. penetrans* spores (blue arrows) and possibly halt in soil when encumbered with approximately 15 *P. penetrans* spores' (Figure 18, red arrow).

All the aforementioned results are graphically presented with generalized linear modeling (Figure 19) using the JASP Statistical Program through the flexplot module. The data presented in Figure 19 show that the gamma (γ) distribution can handle the

'overdispersion' as presented in Table 9, suggesting that the gamma (γ) distribution is probably the most appropriate. As data (counts) presented in Table 9 have values > 0, and exhibit overdispersion (variance > mean), the statistical distribution used to model those counts needs to be calculated and regression with a gamma (γ) distribution seems to be the most appropriate for modeling these counts. The negative binomial cannot provide useful and interpretable information as values (counts) are >0. Using the observations mentioned above, Figure 20 shows that the distribution with data points that roughly follow a straight line fits in a better way the data and appears to have a highest *p*-value, which is the gamma (γ) distribution (*p* = 0.123) followed by Weibull distribution (*p* = 0.083). It is well known that the Weibull and gamma (γ) distributions are commonly used for modeling life systems with monotone failure rates. Physically, this might correspond to a situation in which the object (in our case, the nematode population) fails if a sufficiently large environmental stress occurs (such stresses being distributed according to a Poisson process, as mentioned above in Table 4). In our research, it is reasonable to expect that the soil type with a low sand content, e.g., the loam soil, has the ability to suppress *P. penetrans* attachment. The difference in *P. penetrants* spores' attachment is presented in Table 9. In more detail, the loam soil may indicate that the adhesive properties of these spores are influenced by the soil type, especially the content of the sand. The results presented in this study show the estimation of the *P. penetrans* attachment median for the difference in the number of the two soil types (sandy loam and loam soil): in greater detail, these are the 7 *P. penetrans* spores attached/J2-based on a Mann–Whitney test (Table 10) and 6.25 *P. penetrans* spores attached/J2-based on a Wilcoxon signed rank test—with differences between the population medians of 3–10 and 4.5–8 *P. penetrans* spores, respectively.

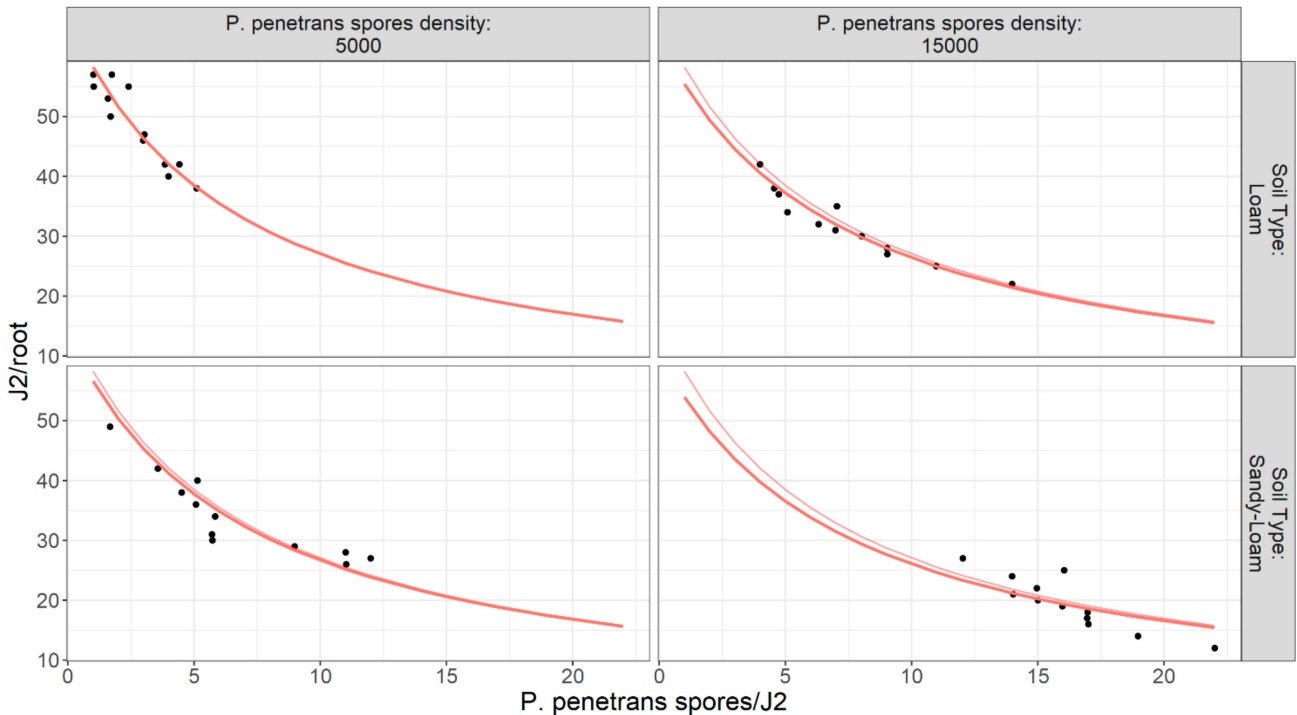

**Figure 19.** Generalized linear modeling plot reflecting the relationship between nematodes invasion (J2s/root) and J2s encumbered with *P. penetrans* spores (*P. penetrans* spores/J2) in two soil types (sandy loam and loam soil). "Ghost Lines" (red narrow line) repeat the pattern from one to the others, making it easier to compare the differences between panels.

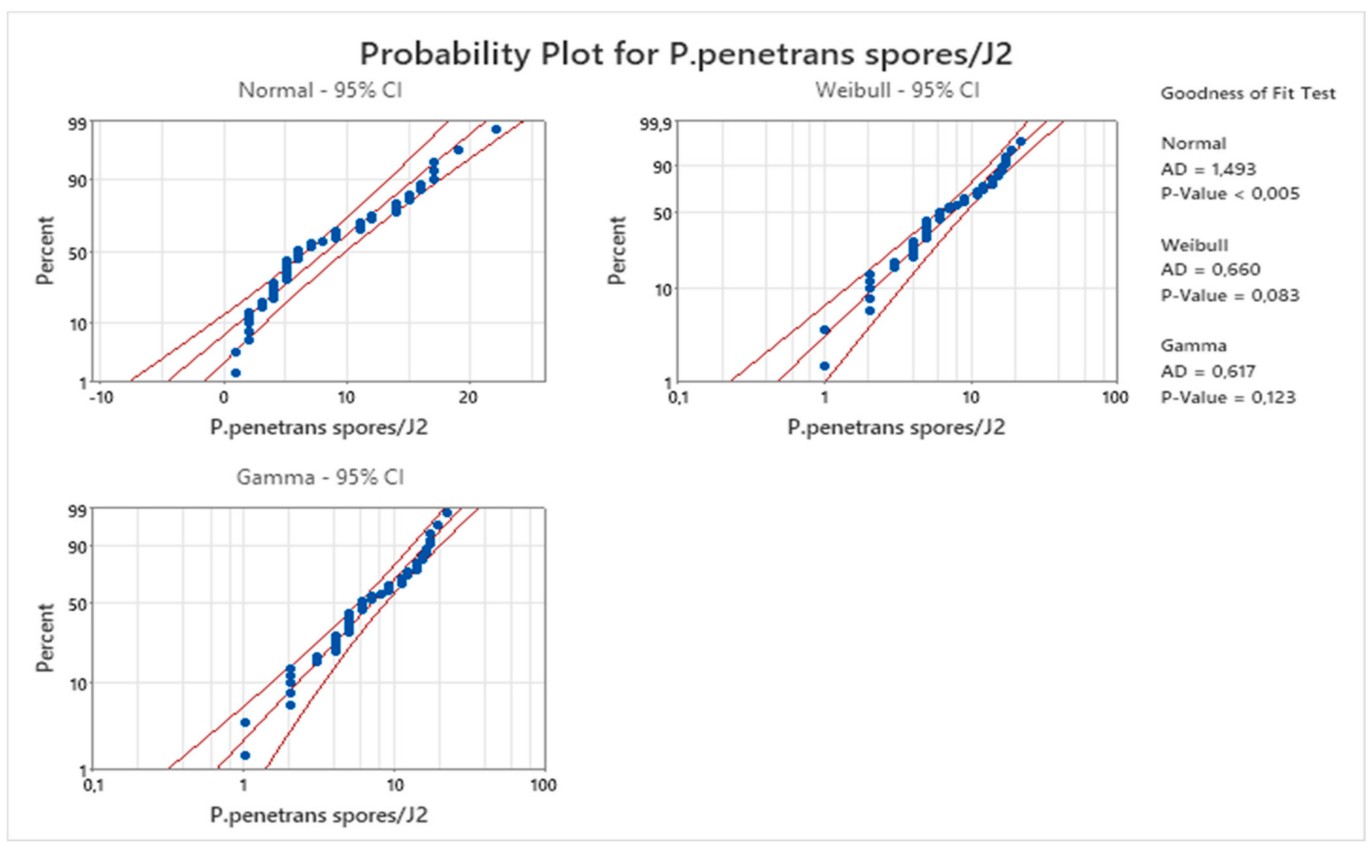

**Figure 20.** Individual distribution identification, e.g., gamma (γ), to choose the distribution that best fits the current data.

**Table 10.** A measure of the central tendencies of *P. penetrans* attachment in sandy loam and loam soil based on a Mann–Whitney test: two-sample rank sum test and Wilcoxon: one-sample signed rank test.

| | | **Mann–Whitney Test** | | | **Wilcoxon Signed Rank Test** | |
|---|---|---|---|---|---|---|
| *n* | **Median** | **CI for Difference** | **Achieved Confidence** | **Median** | **CI for Difference** | **Achieved Confidence** |
| 24 | 7 | (3; 10) | 95.11% | 6.25 | (4.5; 8) | 94.97% |

Finally, it is believed that this study offers strong evidence for solving the optimization problems of the negative binomial and the gamma (γ) distribution using a nonlinear model with four parameters which provides an adequate fit. Figure 21 shows that the fitted line plot follows the observed values which visually indicate that the model fits data for 96.3% R-Sq(adj), (Figure 21). The regression equation is (b1 + b2 × *P. penetrans* spores/J2 + b3 × *P. penetrans* spores/J2^2 + b3 × *P. penetrans* spores/J2^3) where b1 = 64.4; b2 = −7.436; b3 = 0.4774 and b4 = −001151. Table 11 shows the estimates of its four parameters of the nonlinear model.

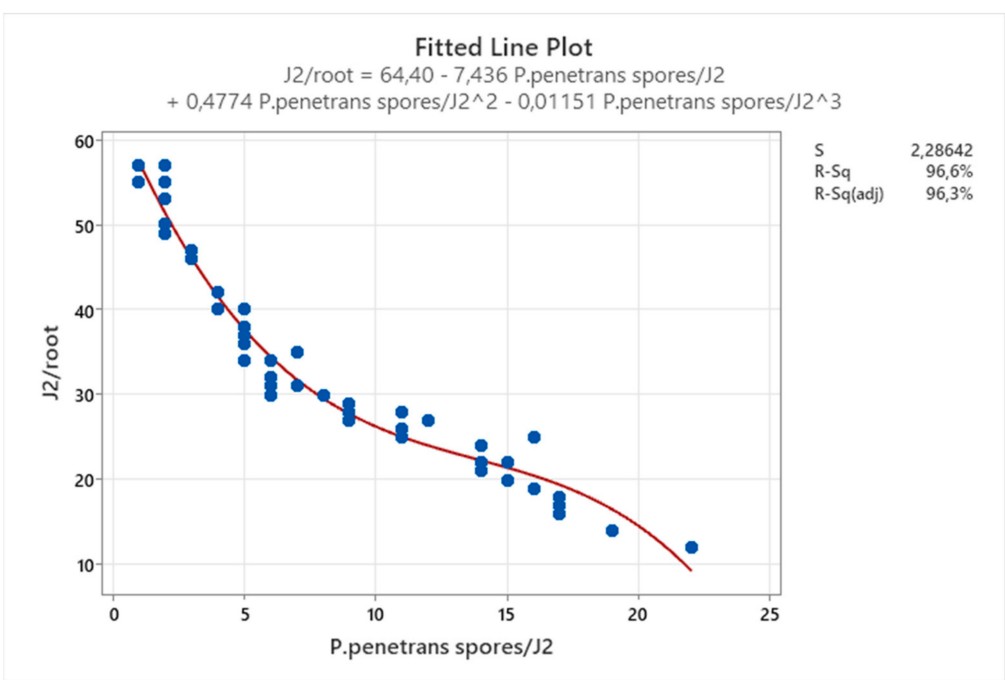

**Figure 21.** Nonlinear regression model which visually indicates that the model fits the data (response(Y): J2/root; predictor(X): *P. penetrans* spores/J2).

**Table 11.** Nonlinear model, parameter estimates.

| Parameter | Estimate | SE Estimate |
|---|---|---|
| b1 | 64.4027 | 1.47605 |
| b2 | −7.4360 | 0.56964 |
| b3 | 0.4774 | 0.05932 |
| b4 | −0.0115 | 0.00180 |

In order to better understand the effect of the attachment of *P. penetrans* spores, the authors of this study used the odds ratio of a binary logistic regression model using the Minitab statistical software where the odds ratio for *P. penetrans* spores/J2 (Table 12) is 2.831. In our binary logistic model, the response was the soil type (loam or sandy loam), the categorical predictor was *P. penetrans* spores' density (5000 or 15,000 spores), and the continuous predictors were J2/root and *P. penetrans* spores/J2.

**Table 12.** Odds for continuous predictors (J2/root and *P. penetrans* spores/J2).

| | Odds Ratio | 95% CI |
|---|---|---|
| J2/root | 0.9728 | (0.6995; 1.3526) |
| *P. penetrans* spores/J2 | 2.8310 | (0.8308; 9.6463) |

## 3. Discussion

Poisson model is usually used to model data. If the observed variability in counts is much larger than the mean, a phenomenon termed overdispersion occurs, and the negative binomial model is often used as a replacement for overdispersed count data.

The application of the Poisson and negative binomial distributions for modeling *P. penetrans* attachment data was also studied by Vagelas et al. [24]. In the aforementioned research [24], evidence is provided of the *P. penetrans* attachment counts being characterized by a significantly larger variance than the mean. This phenomenon in biology is called "overdispersion" [36]. In addition to the aforementioned study [24], the results of the presented study are in agreement with other relevant research studies [23–26], which

concluded that in analyzing *P. penetrans* attachment counts, the variance is significantly larger than the mean, and negative binomial distribution is the most appropriate model to fit the observed data. All previous studies [23–26] which modeled *P. penetrans* overdispersed attachment with data set in water bioassays only provided the simplest answer to the issues that the negative binomial distribution is the only model to explain data variability regarding the overdispersion of *P. penetrans*. This research provided evidence that apart from in the water bioassay, *P. penetrans* overdispersion is related to soil counts suggesting that the negative binomial distribution is the most appropriate model for solving the overdispersion problem.

Overdispersed data can lead to underestimated standard errors and inflated test statistics and in such circumstances, the negative binomial model can be utilized.

In more detail, the data presented in this study were collected from the observed values of *P. penetrans* spore's attachment to nematodes and the predicted values were mainly developed by the Poisson and negative binomial distributions. The *P. penetrans* spore's attachment was modeled at one concentration (5000 spores), in water and the soil bioassay, at four times of exposure (1, 3, 6, and 9 h) and (12, 24, 26, and 48 h), respectively. In the water bioassay, the modeling approaches discussed above confirmed that the Poisson distribution is a sufficient model for nematode populations limited to 1 h of exposure. Interestingly, the mean is equal to the variance suggesting 'under-dispersion'. The same results were presented for the loam soil bioassay, suggesting the 'under-dispersion' phenomenon.

The negative binomial distribution is considered the most appropriate model to fit the data sets for the sandy loam soil and in the water bioassay at 3, 6, and 9 h of exposure. The data show that for sandy loam soil and in a water bioassay, the negative binomial distribution can model 'overdispersed' data sets. It is well known that, in nature, several distributions—which were devised for series in which the variance is significantly higher than the mean—showed that the 'overdispersion' phenomenon is mainly explained by the negative binomial model [34,36–40] or other discrete distributions such as the extended biparametric Waring and the univariate generalized Waring distributions [41–44].

In general, for the inference of the count data, the four most commonly used statistical model distributions are the Poisson, negative binomial, hurdle, and zero-inflated regression models. The hurdle and zero-inflated regression models are used to handle the distribution of the count outcome with excess zeroes, a phenomenon that did not occur in our data. The negative binomial model addresses the issue of overdispersion by including a dispersion parameter that relaxes the presumption of equal mean and variance in the distribution.

In this research, soil data show that, after 36 h of J2s exposure to *P. penetrans* spores, high numbers of *P. penetrans* spores/nematodes were observed and the negative binomial distribution was proposed as the most efficient model for overdispersed data sets, particularly for sandy loam soil. A thorough classification of the negative binomial distribution could be evidence of nematodes encountering clumps of spores. This is true as the odds ratio is 2.831, indicating that these nematodes encounter clumps of nearly three spores, an important result which corrects previous research studies [22,25,26]. Moreover, in the sandy loam soil bioassay, it was demonstrated that those clumps of *P. penetrans* spores affected the nematodes invasion rate.

In this paper, we focused on determining the number of *P. penetrans* attached to J2s cuticle. From this study, it was assumed that the water bioassay observation agrees with the study completed by sandy loam soil, suggesting that when J2s encumbered with >15 spores/juvenile, a possible halt of nematode in the soil could be caused. The same results were reported by Davies et al. [24], where it was commented that when J2s encumbered with >15 spores/juvenile, resulting in a reduction in the invasion level of >70%. The same observation was also mentioned in a previous research study completed by the authors [25,26], where it was shown that the *P. penetrans* spores attached to the nematode cuticle appeared to have a significant impact on nematode's movement (nematode turns, etc.,) which may play a significant role in the nematode distribution and mobility. These results are also observed in the soil. Our data present that when nematodes are encumbered

with 3–10 spores of *P. penetrans* then they are always overdispersed. On the other hand, when nematodes are encumbered with >10 *P. penetrans* spores, the invasion rate is less than 70%, a result which corrects a previous research study [22,26].

In this research, there is evidence that the soil type, (particularly the sandy soil), appears to have a significant impact on *P. penetrans* spore's attachment and thus nematode invasion. Based upon this, significant data are presented suggesting that fewer nematodes invade plant roots when they are encumbered with 3–10 or 4.5–8 *P. penetrans* spores.

This impact of the soil type also significantly affected the nematodes' establishment on the tomato root system when J2s were encumbered with 6–7 *P. penetrans* spores, and possibly immobilized them in the soil when encumbered with >12 or 15 *P. penetrans* spores, especially in the sandy loam soil.

In the examination of the variability of the attachment among sandy loam and loam soil, the negative binomial was not the most appropriate model to fit the loam bioassay data, suggesting an 'under-dispersion' or a probability of a failure [45,46] of *P. penetrans* spores attachment due to the stress environment occurred in the loam soil. A possible explanation caused the low rate of *P. penetrans* attachment at the loam soil may be due to sand contents. Sandy soils may exhibit an increasing rate of *P. penetrans* spores' attachment whereas the absence of sand in loam soil causes attachment failure in clumps of spores. This can be seen as a particular case of our research concluded that the failure rate of attachment, e.g., nematodes being encumbered with fewer than three spores, is characterized by an increasing rate of invasion. At this point, we proposed that the gamma ($\gamma$) distribution can be used for modeling this phenomenon. Gamma ($\gamma$) distribution (as well as Weibull distribution, which was introduced as a second model in our data) may be considered as an under-dispersed extension of Poisson distribution. It is clear that if data exhibit overdispersion, then the variance is larger than the mean. On the other hand, if the data are under-dispersed and if mean = variance = $\lambda$, then the number of events in a given time period of length t follows a Poisson distribution with parameter $\lambda t$. Whereas the Poisson distribution is the answer to this issue, any one step of Poisson mixture distributions assumes that there are only two sources of variability, the negative binomial distribution (which is a Poisson–gamma ($\gamma$) mixture) and the gamma ($\gamma$) distribution [44,47]. Based on that conclusion, it is necessary to mention here that, as the distribution of *P. penetrans* spores attachment is exposed to variance data (which make it of great interest for modeling overdispersed count data sets), a serious draw back to the variance decomposition occurs (as shown in Figures 13 and 14), and it is difficult to determine which model is the most appropriate for referring to data proneness (e.g., the clumps of spores).

The estimates in the *x* and *y* axes of Figures 15 and 16 are of interest as the variance is a multiple of the mean which is consistent with the negative binomial model. The linear regression plots presented with Figures 17 and 18 clearly support (with the "Ghost Line") the hypothesis that the reduction in the invasion level (by approximately 70%) is observed when J2s are encumbered with more than seven (7) *P. penetrans* spores. Moreover, the generalized linear modeling plot in Figure 18 clearly reflects the relationship between nematodes invasion (J2s/root) and J2s encumbered with *P. penetrans* spores (*P. penetrans* spores/J2) in two soil types (sandy loam and loam soil), suggesting that the sand content affects the reduction in the invasion level. Furthermore, the high positive coefficient value of sandy loam soil (6.375) clearly indicates that more *P. penetrans* spores are attached in the sandy loam soil. The coefficient of 6.375 (Table 8) is not only positive but also significant from the coefficient of 0.808 (Table 8), suggesting that the Poisson model is not perfect. Again, the gamma ($\gamma$) distribution (*p* = 0.123) followed by the Weibull distribution (*p* = 0.083) of Figure 20 confirms that, apart from the negative binomial distribution, the gamma ($\gamma$) distribution jointly models the effects of liability (variance) and proneness (a better distribution for the others), especially the coefficient of the sandy loam soil. Thus, the following hypothesis is considered that the reduction in the invasion level (approximately by 70%) is observed when J2s were encumbered with 4.5–8 *P. penetrans* spores as estimated

with the Wilcoxon signed rank test (Table 10) and further confirmed with the cubic line of Figure 21.

## 4. Materials and Methods

### 4.1. Root-Knot Nematode Culture

A culture of root-knot nematodes (*Meloidogyne* spp.) was propagated and maintained on tomato plants (cv. Tiny Tim) cultivated in the glasshouse located at the University of Thessaly facilities. Second-stage juveniles (J2s) were collected from infected tomato roots using the method described by Hussey and Barker [32].

### 4.2. Production of Fresh Second-Stage Juveniles (J2s)

A root-knot nematode culture was maintained on tomato roots cv. Tiny Tim for more than a month. For J2s inoculum, nematode culture plants were uprooted, and infested roots were washed free of soil with tap water. These roots were then cut into small pieces and placed in a jar with 200 mL of a 0.5% sodium hypochlorite (NaOCl) solution and vigorously shaken for 3 min. The use of NaOCl resulted in the dissolution of the gelatinous matrix releasing the eggs from the egg masses [8]. The suspension was passed through 150 and 38 μm mesh sieves in order to collect the root debris in the larger mesh and nematode eggs on the 38 μm sieve. The nematode's eggs were poured onto a tissue paper covering a plastic sieve that was placed in a dish filled with tap water. The dish then was covered with a lid to prevent drying and was further incubated at 28 °C for 2–6 days [16]. In 2–6 days, second-stage juveniles (J2s) were hatched from the eggs and moved through the paper into the water. The hatched J2s were collected every day from 2 to 6 days and these nematodes were used as 'fresh' nematodes for the experiments.

### 4.3. Estimating J2 in the Suspension

The suspension of J2s was poured from the extraction dish into a measuring cylinder and was mixed by blowing with a pipette. After that, five aliquots of 1 mL each were placed in a counting dish and the numbers of J2s were counted under a microscope at a magnification of 35 x. The mean number of J2s of the five aliquots was multiplied with the total volume of suspension to estimate the total juvenile suspension density [25,26].

### 4.4. P. penetrans Culture and Attachment Process

*P. penetrans* strain PpNemJ derived from a commercial product of *P. penetrans* (Pp) Nematech Co., Ltd., Japan, was used in this study.

Attachment studies were carried out to confirm the attachment rate of *P. penetrans* isolation to the fresh J2 root-knot nematodes. Suspensions containing 200 freshly hatched J2s per mL of water were prepared and placed (pipetted) into Petri dishes. A suspension of *P. penetrans* spores, containing approximately 5000 *P. penetrans* spores/mL, was also added into Petri dishes. The Petri dishes were then kept in an incubator with a constant temperature at 28 °C. Nematodes were observed under an inverted microscope at ×200 magnification and the numbers of *P. penetrans* spores attached per nematode from a total of 40 random J2s/treatment were recorded [23–26].

### 4.5. Modeling P. penetrans Spore Attachment Data—Water Bioassay

Spore suspensions of PpNemJ strain were used in this study with sufficient attachment to the *Meloidogyne* spp. as shown by Vagelas et al. [23,25,26]. *P. penetrans* (Pp) spore concentration of 5000 *P. penetrans* spores/mL was used in this study. Endospore's concentrations were prepared from counts made using a hemocytometer and diluted into sterile distilled water as required. *P. penetrans* attachment onto newly hatched J2s of root-knot nematodes was obtained by a centrifugation method [12]. All dishes were placed in a 28 °C incubator. Forty random J2s were observed and the numbers of Pp attached per J2 were recorded under a light microscope. Observations were made at 1, 3, 6, and 9 h after placing nematodes in the spore suspension of 5000 *P. penetrans* spores/mL [15].

Using JASP 0.14.1.0, an open-source statistical program (https://jasp-stats.org/, accessed on 1 December 2021) for Windows, we produced descriptive statistics that summarized the sample distribution in boxplots with violins and jittered data. Furthermore, with JASP statistical software, the best fit discrete distribution was estimated with a probability mass function (PMF), (Equation (1)) or the negative binomial distribution mass function (Equation (2)) before the PMF and the cumulative distribution function (CDF) of the Poisson and negative binomial distribution were displayed with plots. Finally, we used the chi-square test to determine whether a variable is likely to come from a specified distribution (e.g., the Poisson distribution) or not. Where $r$ is the number of successes, $k$ is the number of failures and $p$ is the probability of success.

$$P \ (k \ events \ in \ time \ period) = e - \frac{events}{time} \times time \ period \times \left(\frac{events}{time} \times time \ period\right) k/k! \quad (1)$$

$$f(k; r, p) \ \equiv \ \Pr(X = k) = \binom{k + r - 1}{r - 1} (1 - p)^k \ p^r \quad (2)$$

*4.6. Estimating Root-Knot Nematodes Development via Water Bioassay*

Sets of treatments of fresh J2s of root-knot nematodes that were exposed or not to 5000 spores per Petri dish in water were used to estimate the nematode development on tomato roots. Fresh J2s were exposed to *P. penetrans* spores as described above (2.5.1.) and observations of the attachment were made under an inverted microscope at ×200 at 9 and 36 h after placing the nematodes in spore suspension. Furthermore, root-knot nematode gall and egg masses were produced on test tomato plants, plastic pots in a glasshouse experiment. Pots with tomato plants were treated with J2s that were exposed or not to 5000 *P. penetrans* spores for 9 h and kept in the glasshouse (temperature: $25 \pm 4$ °C) for 32 days. After 32 days, infested roots were excised from stems, drained of excess water, and pressed to uniform dryness. Roots were then observed under a stereoscope at ×25–30 magnification for gall and egg masses. These assessments (plant gall and nematode egg mass) were subjected to descriptive statistics using the JASP statistical software version 0.14.1.0 and summarized the sample distribution in boxplots with violins and jittered data. The mean, variance, and discrete distribution were performed using JASP statistical software. Each treatment had 40 replicates (Petri dishes bioassay) and 12 replicates (pots bioassay). Moreover, a flexplot utility was used in order to visualize multivariate relationships between gall and egg masses. The difference between treatments was compared with a "Ghost Line". "Ghost Line" repeats the relationship from one panel, e.g., Figure 8, J2s treated with 6–11 *P. penetrans* spores, to the other panel, e.g., Figure 8, J2s treated without or with 24–32 *P. penetrans* spores, to facilitate the comparison of the relationship or differences.

*4.7. Modeling P. penetrans Spore Attachment Data-Soil Bioassay*

Sets of treatments in water suspension as described above (2.5.1.) were made in sandy loam and loam soil by adding a small amount (1 g) of sterile sandy loam or loam soil. In detail, 1 g of soil was mixed with 3 mL of tap water to make the soil bioassay. For attachment in the soil bioassay, 'fresh' J2s of root-knot nematodes were exposed to 0, 5000, and 15,000 spores per ml and a total of 60 random nematodes were examined for *P. penetrans* spore attachment from each treatment after the incubation of the Petri dishes at 28 °C for 36 h.

*4.8. Estimating J2s Root Invasion Rate Encumbered with Different Doses of P. penetrans Spores in Sandy Loam and Loam Soil*

Sets of treatments for the attachment bioassay of fresh J2s of root-knot nematodes that were exposed to 5000 and 15,000 spores per Petri dish in sandy loam and in loam soil were used in order to estimate the J2s invasion rate. Treatments were: (1) tomato plants in sandy loam soil treated only with J2s; (2) tomato plants in loam soil treated only

with J2s; (3) tomato plants in sandy loam treated with 5000 *P. penetrans* spores/mL in sandy loam bioassay for 36 h as described above; (4) tomato plants in sandy loam treated with 5000 *P. penetrans* spores/mL in sandy loam bioassay for 36 h; (5) tomato plants in sandy loam treated with 15,000 *P. penetrans* spores/mL in sandy loam bioassay for 36 h; (6) tomato plants in loam soil treated with 5000 Pp spores/mL in loam bioassay for 36 h; and (7) tomato plants in loam soil treated with 15,000 *P. penetrans* spores/mL in a loam bioassay for 36 h. Each treatment was replicated twelve times. The number of J2s per root system was determined 12 days after inoculation following the Hooper [33] method. For all pot treatments, plants were inoculated with 600 fresh J2s.

Using the JASP statistical software version 0.14.1.0, the sample distribution was summarized in boxplots with violins and jittered data, as described above (2.5.1). Furthermore, the interactions between soil type (sandy loam and loam soil) and *P. penetrans* spores' density (5000 and 15,000/mL) were modeled with ANOVA and Linear Models.

Finally, from the sandy loam and loam soil bioassay, after 36 h of incubation, ten random nematodes encumbered with 5000 or 15,000 *P. penetrans* spores/mL were examined for *P. penetrans* attachment under ×200 magnification with an inverted microscope. These data were used to determine the relationship between encumbered nematode and invasion rate at the two tested soil types (sandy loam and loam soil).

## 5. Conclusions

This paper is concerned with fitting discrete distribution models to *P. penetrans* attachment in water and soil bioassay tests. Those attachments were further tested for nematode invasion in planta. This research study concludes that the number of spores attached in J2s, the time of exposure of J2s to *P. penetrans* spores, and the soil texture are important factors affecting the invasion of root-knot nematodes in tomato plants.

**Author Contributions:** This research is a product of the intellectual environment of all authors which contributed in various degrees to the analytical methods used, the research concept, and the experiment design. I.V. designed the study, developed the methodology, and performed the analysis, while S.L. and I.V. collected the data. I.V. and S.L. also contributed to writing the manuscript. All authors have read and agreed to the published version of the manuscript.

**Funding:** This research received no external funding.

**Institutional Review Board Statement:** Not applicable.

**Conflicts of Interest:** The authors declare that they have no conflict of interest.

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
