# Peer review of "Modeling the Overdispersion of Pasteuria penetrans Endospores"

_parasitologia, doi:10.3390/parasitologia2030018_

Round 1

Reviewer 1 Report

The manuscript has publication merit.

Congratulations to the authors for the excellent article

The article needs improvement mainly in the figures, as some scientific names are not in italics.

Author Response

REVIEWER 1

The manuscript has publication merit.

Congratulations to the authors for the excellent article

Reviewer 1 Comments

Corrections

The article needs improvement mainly in the figures, as some scientific names are not in italics.

Dear reviewer, we thank you for the observation. The style within the text was corrected according to your observations. Unfortunately, the style is not easy to be corrected in some Figures.

Reviewer 2 Report

The style of writting is not full understandable to readers which do not have a strong background on statistics. The Abstract does not cover all aspects of the paper. Several Figures are not mentioned in the Results (e.g Figure 1, 4 etc). Some terms are not common to readers who do not have a strong background on statistics e.g Ghost Line, violins, jittered data etc. What does mean that spores attach or not attach in clumps? They attach individually on juveniles and not in clumps or clusters. Was exclusively the commercial product of Pasteuria penetrans used? Or was it cultured by the authors? How are the spores formulated in the commercial product? In lines 509-510, it is not clear whether the authors used or not the centrifugation method for spore attachment. It is mentioned that replicates were three folds and 40 J2s were observed. Does in mean a total of 40J2s for each replicate, 120J2s per treatment, or 40J2s in total, e.g 13J2s per replicate? The soil bioassay is not clear (lines 549-554). The J2s were added to 1g of soil and inculated for certain period. How were these J2s isolated from soil for microscope observation?In lines 560-564, the treatments 3 and 4 appear as being the same. Was an inverted microscope used for J2 observation? Or were the juveniles put in microscope slides and observed in a light microscope? The authors do not refer in any study on Pasteuria penetrans attachment in different soil types. Are they sure that there is not such a work published?

The manuscript has several flaws and  requires substantial revision by the authors. It is not clear and understandable to readers with no special background on statistics.

Author Response

REVIEWER  2

Reviewer 2 Comments

Corrections

1.       The Abstract does not cover all aspects of the paper

Dear reviewer, we thank you for the observations. A small phrase is added in abstract section.

2.       Several Figures are not mentioned in the Results (e.g Figure 1, 4 etc).

All figures have been checked. Figure 1 is added. Figure 4(in line 117) and the rest of the Figures are previously mentioned.

3.       Some terms are not common to readers who do not have a strong background on statistics e.g Ghost Line, violins, jittered data etc.

The main subject of the manuscript is nematodes and statistic. The first author holds a second degree (MPhil) in statistics modelling. The readers must have a strong background in statistics in order to understand the findings of the manuscript

4.       What does mean that spores attach or not attach in clumps? They attach individually on juveniles and not in clumps or clusters.

Probably individually not in clumps

5.       Was exclusively the commercial product of Pasteuria penetrans used? Or was it cultured by the authors?

They were cultured by the authors

6.       How are the spores formulated in the commercial product?

As granules

7.       In lines 509-510, it is not clear whether the authors used or not the centrifugation method for spore attachment.

There is a centrifugation method. Line 539.

8.       It is mentioned that replicates were three folds and 40 J2s were observed. Does in mean a total of 40J2s for each replicate, 120J2s per treatment, or 40J2s in total, e.g 13J2s per replicate?

Replicates in three folds were mentioned by mistake. Total 40 nematodes. Line 540

9.       The soil bioassay is not clear (lines 549-554). The J2s were added to 1g of soil and inοculated for certain period. How were these J2s isolated from soil for microscope observation?

J2s were isolated from soil for microscope observation by dilution in tap water

10.    In lines 560-564, the treatments 3 and 4 appear as being the same. Was an inverted microscope used for J2 observation? Or were the juveniles put in microscope slides and observed in a light microscope?

An inverted microscope was used for the observations.

11.    The authors do not refer in any study on Pasteuria penetrans attachment in different soil types. Are they sure that there is not such a work published?

So far, from our knowledge there is no similar research work in soil and water conditions.

12.    The manuscript has several flaws and requires substantial revision by the authors. It is not clear and understandable to readers with no special background on statistics.

The readers must have a strong background in statistics in order to understand the findings of the manuscript.

Reviewer 3 Report

The article is interesting, but it needs to be improved in several ways:

-The English needs to be improved. I am not an English speaker person, but it is complicated to follow and a person expert in this subject is needed.

-The editing of the paper need to be improved, exemples as "J2s are attached by the...is "J2s are attracted by the...", other as missing letters in words, italics in species names, many crossed out words are found in the document,  etc,... 

-Fig.1 legend: define JASP.

-Fig. 9 legend: four P. penetrans densities?

-Results section needs to be simplified,...it is hard to follow in my opinion, as I am a nematologist, not an expert in statistics.

-Results are mixed with discussion (i.e. l292-298, and others)

Author Response

REVIEWER  3

Reviewer 3 Comments

Corrections

1.       The English needs to be improved. I am not an English speaker person, but it is complicated to follow and a person expert in this subject is needed.

There was an effort in order to improve English level of the manuscript

2.     The editing of the paper need to be improved, exemples as "J2s are attached by the...is "J2s are attracted by the...", other as missing letters in words, italics in species names, many crossed out words are found in the document,  etc,... 

We edit the manuscript. Some minor corrections are made.

3.       Fig.1 legend: define JASP.

open-source statistical program

4.       Fig. 9 legend: four P. penetrans densities?

It was for and not four (it was written by mistake)

5.     Results section needs to be simplified,...it is hard to follow in my opinion, as I am a nematologist, not an expert in statistics.

Unfortunately, readers must have a strong background bot only in nematology but also in statistics in order to understand the findings of the manuscript

6.       Results are mixed with discussion (i.e. l292-298, and others)

Dear author we made an effort to separate discussion and results section in previous submission.

Round 2

Reviewer 2 Report

I consider that the authors could give some explanations on the statistical terms. Accordingly to my opinion, this article with some improvement, could become clear to readers without a strong background on statistics. It looks likely that the authors  insist that the manuscript is attented exclussively to readers with background on Pp - RKN interaction  and a strong background on statistics.  Therefore, I can not recommend publication.

Author Response

We would like to thank the reviewers for the efforts they made to improve this article. We made all the proposed from reviewers’ corrections and we are open to correct all other next possible observations. Please check our responds in the observations.

REVIEWER 2

Reviewer 2 Comments

Corrections

I consider that the authors could give some explanations on the statistical terms. Accordingly to my opinion, this article with some improvement, could become clear to readers without a strong background on statistics. It looks likely that the authors insist that the manuscript is attented exclussively to readers with background on Pp - RKN interaction and a strong background on statistics.  Therefore, I can not recommend publication.

Small sentences (yellow marked) that will help the readers has been added to the text.

Reviewer 3 Report

The article is fine with the corrections performed.

Author Response

We would like to thank the reviewers for the efforts they made to improve this article. We made all the proposed from reviewers’ corrections and we are open to correct all other next possible observations. Please check our responds in the observations.

REVIEWER  3

Reviewer 3 Comments

Corrections

The article is fine with the corrections performed.

Thank you for your suggestions

Round 3

Reviewer 2 Report

The manuscript is not clear to readers without a strong background on statistics. The authors have not explained efficiently many statistical terms.